# High Speed Microactuators for Low Aspect Ratio High Speed Micro Aircraft Surfaces

**Ronald Barrett-Gonzalez * and Nathan Wolf**

Aerospace Engineering Department, University of Kansas, Lawrence, KS 66044, USA; nathan.w@ku.edu
* Correspondence: barrettr@ku.edu; Tel.: +1-785-864-2226

**Abstract:** This paper covers a class of actuators for modern high speed, high performance subscale aircraft. The paper starts with an explanation of the challenges faced by micro aircraft, including low power, extremely tight volume constraints, and high actuator bandwidth requirements. A survey of suitable actuators and actuator materials demonstrates that several classes of piezoceramic actuators are ideally matched to the operational environment. While conventional, linear actuation of piezoelectric actuators can achieve some results, dramatic improvements via reverse-biased spring mechanisms can boost performance and actuator envelopes by nearly an order of magnitude. Among the highest performance, low weight configurations are post-buckled precompressed (PBP) actuator arrangements. Analytical models display large deflections at bandwidths compatible with micro aircraft flight control speed requirements. Bench testing of an example PBP micro actuator powered low aspect ratio flight control surface displays $+/-11°$ deflections through 40 Hz, with no occupation of volume within the aircraft fuselage and good correlation between theory and experiment. A wind tunnel model of an example high speed micro aircraft was fabricated along with low aspect ratio PBP flight control surfaces, demonstrating stable deflection characteristics with increasing speed and actuator bandwidths so high that all major aeromechanical modes could be easily controlled. A new way to control such a PBP stabilator with a Limit Dynamic Driver is found to greatly expand the dynamic range of the stabilator, boosting the dynamic response of the stabilator by more than a factor of four with position feedback system engaged.

**Keywords:** micro flight control; high speed; piezoelectric; low aspect ratio

## 1. Introduction

New technologies are bringing new challenges to aircraft of many scales. While flight control actuators for large, inhabited aircraft have been around since Sperry's autopilot first took to the air more than a century ago, small flight control actuators are still challenged [1]. One of the most daunting challenges is related to low actuator bandwidths. Figure 1 is a Bode amplitude plot of several conventional subscale flight control actuators.

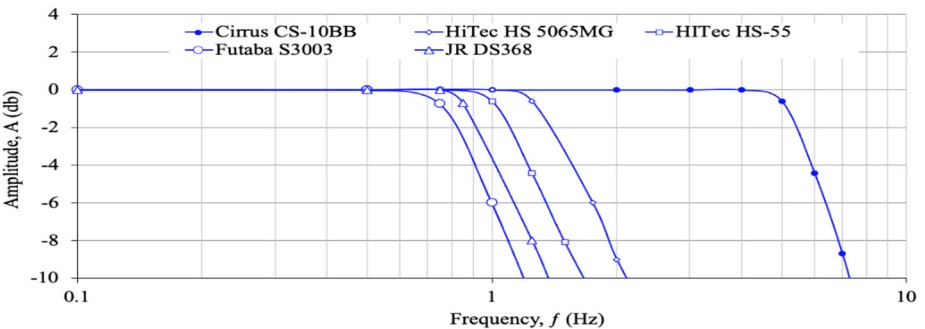

**Figure 1.** Bode Amplitude Plot of five Common Sub-Submicro Scale Actuators [2].

If one examines the aeromechanics of many subscale aircraft, it is easy to see that for very slow aircraft, such as model airplanes flying far away from any structures in an open field, the aforementioned actuators will work well. Indeed, the vast majority of subscale aircraft actuators are built for toy airplanes and helicopters operating in environments like this. As subscale aircraft move faster and faster and/or enter ever more challenging flight dynamics environments, greater and greater speeds are needed. As part of a series of surveys performed in the 1990s supporting the DoD Micro Aerial Vehicle (MAV) program, detailed surveys and studies of gust structures and intensities were recorded in and around an urban environment on a 50% atmospherics day [3]. The results displayed below demonstrate intense upset angles and frequencies for aircraft operating in urban environments.

Clearly, upset gusts on the order of 10 kts peak-to-peak at 10 Hz are enough to tax many flight control actuators. If an aircraft is flying at an airspeed of just 18kts, a 10 kt upset gust magnitude leads to a 30° upset angle, which is beyond the stall angle for many aerodynamic surfaces. These atmospheric structures are the result of airflow patterns like building rollers and street devils, which tend to be relatively violent, localized, and separated vortices that sweep through urban centers and around buildings. While the casual observer can view leaves and trash carried aloft in spiral structures, such items can spell doom for subscale aircraft that try to fly through them. These structures also occur in and around mountains, hills, and valleys. In general, the larger the geographic feature, the longer period/slower frequency of the gust structure. While maintenance of flightworthiness is a primary concern, large amplitude and/or high frequency gusts are problematic in other ways. The most common problems are associated with blurred images as airframes are shaken. Because some UAVs are built as point interceptors, high frequency and magnitude gust fields adversely impact their ability to guide the aircraft to intercept.

To control aircraft in challenging gust fields, pitch, roll, and yaw motions must also be controlled actively. Low aspect ratio flight control surfaces are most often found controlling longitudinal and directional modes (e.g., pitch and yaw). Low aspect ratio flight control surfaces are typically used in empennages and, as such, the fastest aeromechanical modes they usually control are short period longitudinal modes wherein the natural frequency of the short period mode is the fastest frequency that the flight control system deals with [4]:

$$\omega_{nsp} = \sqrt{\frac{Z_\alpha M_q}{U_1} - M_\alpha} \tag{1}$$

As one examines a variety of aircraft, it is easy to see that full scale and subscale short period frequencies on a 30 cm (1 ft) scale are dramatically different. By using the aeromechanics listed in [4] of a variety of aircraft, the short period modes are much greater, especially if flight speeds are held constant.

By examining the actuator data of [2] and in Figure 1, one observes the genesis of a nontrivial problem: to control subscale aircraft moving at high speeds and suppress natural aeromechanical modes, the flight control actuators must be very fast, certainly faster than those used by model airplanes as seen in Figure 2. It is also interesting to note that, from Figure 3, the excitation frequencies of atmospherics where many subscale aircraft are expected to fly are precisely matching short period modes. Typically, long period or phugoid modes are about an order of magnitude slower, which also leads to problems, as the atmospherics of Figure 3 display even larger upset magnitudes at lower frequencies. Clearly, subscale high performance aircraft have some daunting aeromechanic issues to deal with.

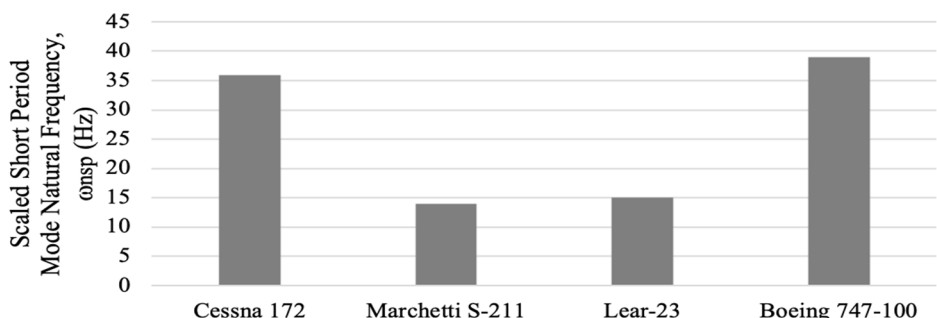

**Figure 2.** Short Period Modes of Aircraft Scaled to a 30 cm (1 ft) Size.

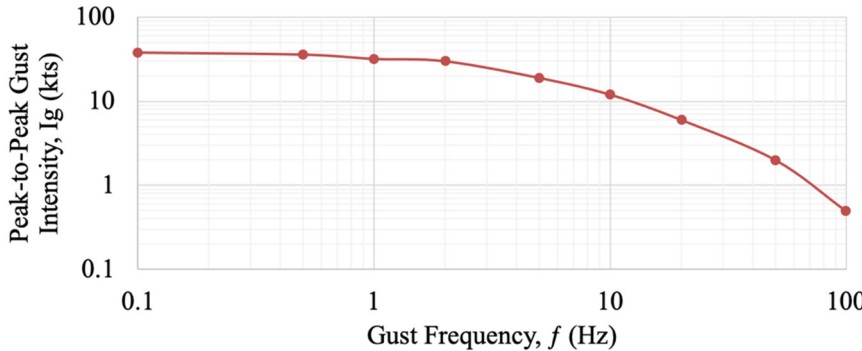

**Figure 3.** Typical Urban Maximum Gust Profile, 50% Far Field Day.

Another major issue confronting subscale high performance aircraft is related to volume constraints. Every cc of volume occupied within a fuselage of high performance drones comes at a premium. Some systems like the AeroVironment Switchblade UAV are representative of the struggles faced by subscale aircraft design engineers today. At more than $70,000 each, the volumetric cost is at least $25/cc [5]. The gross weight and cost sensitivity is nearly double that since volume drives wetted area, which in turn increases parasite area. Including cost sensitivities, using the techniques of [6], volume alone drives cost by as much as $47/cc. Accordingly, there is a strong drive to eliminate as much volume as possible from high performance UAVs. To accomplish this, one technique is to push the actuators out of the fuselage completely and into flight control surfaces. Given the form factors of conventional actuators referred to in Figure 1, such a move is fundamentally not possible. Also driving actuator volume is power consumption. The actuators of Figure 1 consume as much as 12 W when driving control surfaces at high speeds and high deflections; this in turn increases driving electronics volume, weight, and cost.

The final issues of note are related to characteristics that very adversely affects pointing accuracy: stiction, friction, and slop. Testing a range of subscale microactautors revealed nontrivial problems with conventional installations, as displayed in Figure 4.

Because many subscale high performance aircraft are designed to be highly maneuverable, they often possess trimmed control ratios in excess of $\partial \alpha / \partial \delta_e > 1$ and static margins that are less than 5% MGC. Accordingly, small amounts of errors, as found above, can lead to large angle of attack and therefore pointing errors. Such pointing errors can cause relatively large miss distances in the case of pursuit aircraft or lead to sensor jitter or pointing problems.

Thankfully, there are new classes of flight control actuators and actuator materials available. While many actuator classes and materials are notoriously slow and often bulky with unfavorable form factors, a relatively fast class has matured over the past three decades. The earliest adaptive flight controls using piezoelectric materials were investigated by Crawley's lab at MIT in the late 1980s [7–11]. They demonstrated that

plates could be actively cambered and twisted by using various kinds of coupling, flaps could be deflected, and small deflections could be imparted to rotor blades. By mounting piezoelectric elements of various configurations, small strains of only a few hundred microstrain could generate a few percent of camber deflection or fractions of a degree of twist. A series of bench top tests demonstrated that plate twist could be used to generate nontrivial air loads under the right conditions.

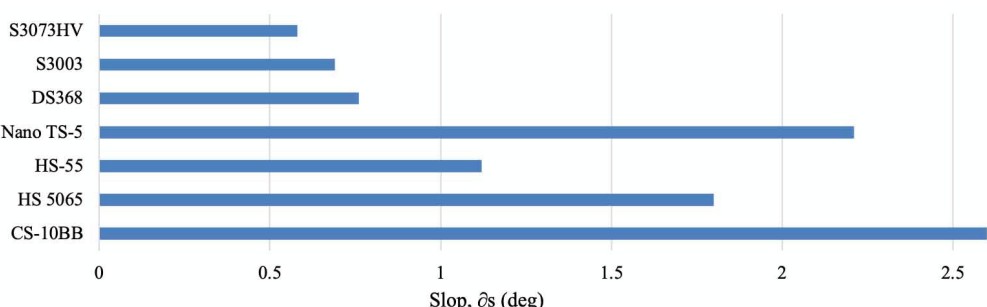

**Figure 4.** Conventional Subscale Servoactuator Slop.

The first all-moving piezoelectric flight control surfaces to be made just after these early experiments were for subsonic missiles [12–18]. These early devices could generate a few degrees of pitch deflections by a variety of mechanisms. One of the first pitch-active devices used a piezoelectric torque-plate that was connected at the tip of the flight control surface. As the plate would twist, the surface would be moved in pitch. Other mechanisms used laterally placed piezoelectric bender elements to move aeroshells. The first attempt at favorable aeroelastic coupling was made in 1995–1996, as a symmetrical subsonic aeroshell elastic axis was displaced afterward with respect to the aerodynamic center. This formed a reverse-bias spring which amplified deflections with increasing airspeed [19]. Although this was perhaps the first reverse-bias spring arrangement used in a piezoelectric flight control surface, the true innovation in reverse-bias spring mechanisms came in 1997 when Lesieutre and his team discovered that axial loads on piezoelectric bimorph benders could amplify deflections, driving electrical-to-mechanical efficiencies close to 100% [20,21]. By building upon the reverse bias configurations of [20,21], these near ideally efficient post-buckled precompressed (PBP) actuators were integrated into a variety of flight control surfaces [22–29]. Many of them flew on a variety of aircraft. Since these highly efficient PBP actuators and associated surfaces were made, a number of other adaptive actuator configurations have come along, most taking the form of wing "morphing" actuators [30,31].

## 2. Preferred Materials and Actuator Configuration

There are many different families of conventional and adaptive flight control actuators, but given that the PBP actuator configurations exhibit the highest efficiency of all adaptive (and conventional) actuators and have outstanding bandwidths with no friction, stiction, or slop, they are natural choices for low aspect ratio flight control surfaces with demanding efficiency, weight, power, accuracy, and speed requirements. By using the techniques first developed by Lesieutre and his team, it was found more than two decades ago that transfer efficiencies can asymptotically approach 100%. For a single cycle of actuation, it is even possible to develop actuation efficiencies that go beyond 100% in the form of snap-through or "mousetrap" configuration actuators. Such single-cycle/triggered actuators are usable in release mechanisms.

Figure 5 dsiplays clearly that two branches of piezoelectric materials, the lead zirconate niobate (PZN) and lead zirconate titanate (PZT) lines, are conducive to being used in a PBP configuration and therefore can possess electrical-to-mechanical conversion efficiencies of just under 100%. While PZN is clearly superior in energy density with respect to PZT, material cost and driving voltages are often prohibitive. PZT in a PBP configuration also

achieves superior electrical-to-mechanical work efficiency while costing substantially less under driving fields that are roughly an order of magnitude lower than PZN. This is an important characteristic as minimization of power consumption, operational voltages, and costs are critical in reduction of vehicle weight, volume, and cost. While just using a simple PBP configuration clearly provides benefits (as found in [22–28]), for high speed flight control a modification of these earlier approaches is needed. Biomimetic adaptive actuator classes are shown to have excellent mass specific energy and transfer efficiencies as seen in [32]. Ref. [33] includes some of the latest actuation schemes in addition to PBP amplification. If one examines a typical PBP actuator element, it is easy to observe that axial force applied to the end at levels around the perfect long column buckling limit tends to maximize deflections. Figure 6 displays a very basic PBP actuator assembly with a precompression spring on the actuator end [34].

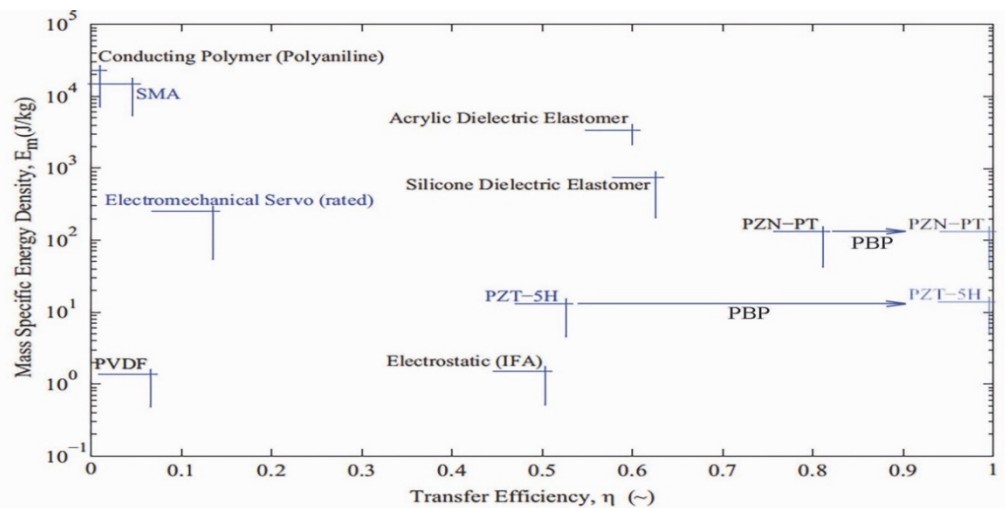

**Figure 5.** Primary Flight Control Actuator Classes and Transfer Efficiencies [22–28,32].

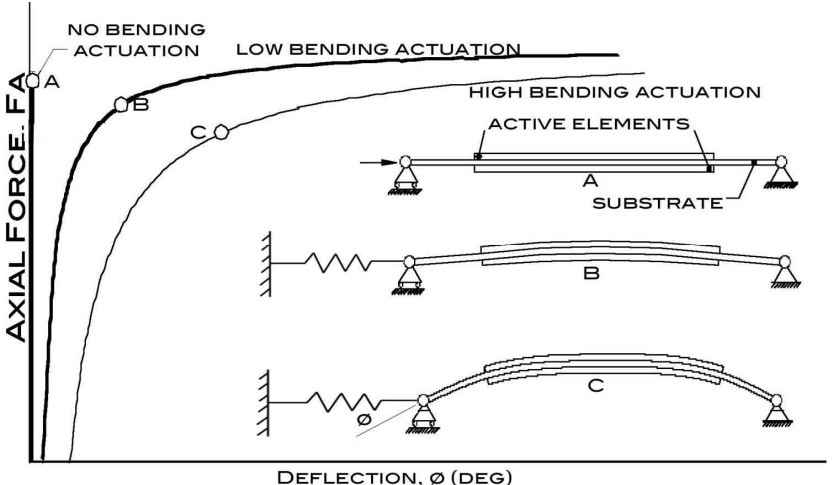

**Figure 6.** Basic Post-Buckled Precompressed Actuator Configuration under Low Axial Force (A), Moderate Axial Force (B), and High Axial Force (C) [34].

While the PBP actuator configuration has been successfully used in production aircraft like the XQ-138 convertible coleopter, it is still relatively unknown among aerospace actuator types and classes in spite of its superior performance.

### 3. Subscale Actuator Configuration, Flight Control Surface Integration, and Analysis

If one examines the PBP work of the early 2000s ([22–28]), nearly all were controlled by tying an outer feedback loop around deflection, ø via a rotational transducer. While this approach does well in controlling gross deflections and has been used in fielded aircraft like the XQ-138, feedback from the surface of the actuator element itself provides lag-free feedback. A new configuration of feedback controlled PBP Flexspar actuator was conceived and reduced to practice. This configuration uses PBP-amplified cantilevered piezoelectric benders to drive an aerodynamic shell in pitch. Figure 7 demonstrates the overall configuration of the actuator and its major components. This 5 cm semispan × 2.5 cm chord low aspect ratio flight control surface is mounted to a main spar constructed of surgical grade stainless steel tubing and collocated with the line of aerodynamic centers at the airfoil quarter-chord. A precompression band places the actuator in longitudinal compression with a quiescent axial force so high that the actuator is unable to stay in the undeformed/undeflected position without feedback engaged. A single-ply graphite-epoxy aeroshell structurally accommodates the leading edge counterbalance and is attached so as to rotate freely about the main spar via a slip-bushing. The total shell deflection is achieved by rigid-body pitch deflections, as displayed in Figure 7 below.

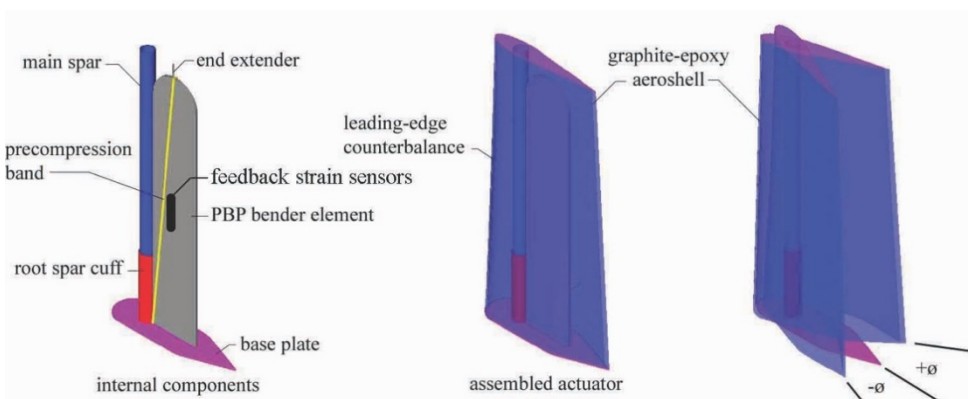

**Figure 7.** PBP Actuator Configuration within 5 cm × 2.5 cm Aeroshell.

One of the challenges that a properly designed PBP system encounters is that its passive natural frequency is driven closer and closer to zero as the net effective stiffness of the actuator system is driven to zero. If one examines the actuator element alone, then there exists a relationship between the axial forces and baseline deflections. If the lengths along the actuator and deflection components are carefully laid out, a closed-form solution can be had using the general actuator configuration as shown below in Figure 8.

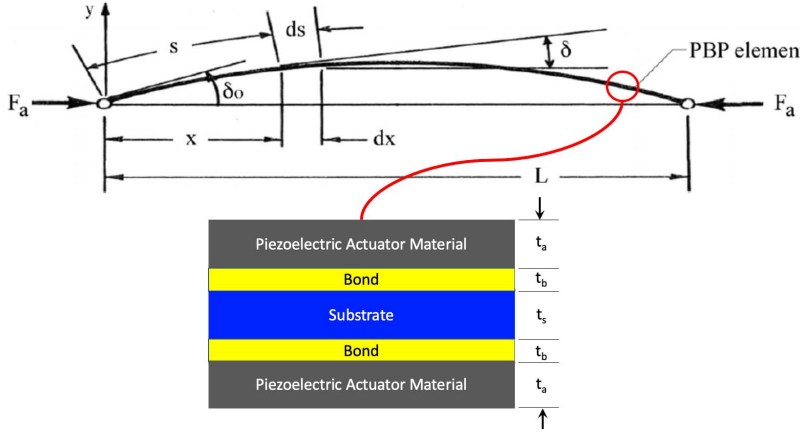

**Figure 8.** PBP Element Solution Conventions for Closed-Form Deflection Solution.

By using standard laminate plate theory as recited in [35], the unloaded circular arc bending rate $\kappa_{11}$ can be calculated as a function of the actuator, bond, and substrate thicknesses ($t_a$, $t_b$, and $t_s$, respectively) and the stiffnesses of the actuator $E_a$ and substrate $E_s$ (assuming the bond does not participate substantially to the overall bending stiffness of the laminate). As driving fields generate higher and higher bending levels of a symmetric, isotropic, balanced laminate, the unloaded, open-loop curvature is as follows:

$$\kappa_{11} = \frac{E_a\left(t_s t_a + 2t_b t_a + t_a^2\right)\Lambda_1}{\frac{E_s t_s^3}{12} + E_a\left[\frac{t_a(t_s + 2t_b)^2}{2} + t_a^2(t_s + 2t_b) + \frac{2}{3}t_a^3\right]} \tag{2}$$

By manipulating the input field strengths over the piezoelectric elements, different values for open-loop strain, $\Lambda_1$ can be generated. This is the primary control input generated by the flight control system (typically delivered by voltage amplification electronics). To connect the curvature, $\kappa_{11}$ to end rotation, and then shell deflection, one can examine the strain field within the PBP element itself. If one considers the normal strain of any point in the PBP element at a given distance, y from the midpoint of the laminate, then the following relationship can be found:

$$\varepsilon = \frac{y\,d\delta}{ds} = \frac{\sigma}{E} \tag{3}$$

By assuming that the PBP beam element is in pure bending, then the local stress as a function of through-thickness distance is as follows:

$$\sigma = \frac{My}{I} \tag{4}$$

If Equations (3) and (4) are combined with the laminated plate theory conventions of [35], then the following can be found, counting $D_l$ as the laminate bending stiffness:

$$\frac{y\,d\delta}{ds} = \frac{My}{D_l b} \tag{5}$$

The moment applied to each section of the PBP beam is a direct function of the applied axial force $F_a$ and the offset distance, $y$:

$$M = -F_a y \tag{6}$$

Substituting Equation (6) into (5) yields the following expression for deflection with distance along the beam:

$$\frac{d\delta}{ds} = \frac{-F_a y}{D_l b} \tag{7}$$

Differentiating Equation (7), with respect to the distance along the beam, yields:

$$\frac{d^2\delta}{ds^2} = -\frac{F_a}{D_l b}\sin\delta \tag{8}$$

Multiplying through by an integration factor allows for a solution in terms of trig. functions:

$$\frac{d\delta}{ds}\frac{d^2\delta}{ds^2} = -\frac{F_a}{D_l b}\sin\delta\frac{d\delta}{ds} \tag{9}$$

Integrating Equation (9) along the length of the beam dimension s yields:

$$\left(\frac{d\delta}{ds}\right)^2 = 2\frac{F_a}{D_l b}\cos\delta\frac{d\delta}{ds} + a \tag{10}$$

From Equation (2), the curvature ($\kappa_{11}$) can be considered a curvature "imperfection", which acts as a triggering event to initiate curvatures. The larger the applied field strength across the piezoelectric element, the greater the strain levels ($\lambda_1$), which results in higher imperfections ($\kappa_{11}$). When one considers the boundary conditions at $x = 0$, $\delta = \delta_o$. Assuming that the moment applied at the root is negligible, then the curvature rate is constant and equal to the laminated plate theory solution: $d\delta/ds = \kappa_{11} = \kappa$. Accordingly, Equation (10) can be solved given the boundary conditions:

$$a = 2\frac{F_a}{D_l b}(cos\delta - cos\delta_0) + \kappa^2 \tag{11}$$

Making proper substitutions and considering the negative root because the curvature is negative by prescribed convention:

$$\frac{d\delta}{ds} = -2\sqrt{\frac{F_a}{D_l b}}\sqrt{\left(sin^2\left(\frac{\delta_0}{2}\right) - sin^2\left(\frac{\delta}{2}\right)\right) + \frac{\kappa^2 D_l b}{4F_a}} \tag{12}$$

For a solution, a simple change of variable aids the process:

$$sin\left(\frac{\delta}{2}\right) = csin\xi \tag{13}$$

The variable $\xi$ takes the value of $\pi/2$ as $x = 0$ and the value of 0 at $x = L/2$. Solving for these bounding conditions yields:

$$c = sin\left(\frac{\delta_0}{2}\right) \tag{14}$$

Making the appropriate substitutions to solve for deflection ($\delta$) along the length, then differentiating yields:

$$\delta = 2sin^{-1}\left(sin\left(\frac{\delta_0}{2}\right)sin\xi\right) \quad \ni \quad d\delta = \frac{2sin\left(\frac{\delta_0}{2}\right)cos\xi}{\sqrt{1 - sin^2\left(\frac{\delta_0}{2}\right)sin^2\xi}}d\xi \tag{15}$$

Combining Equations (12)–(15) and solving for the appropriate boundary conditions yields the following closed-form solution relating deflections and axial forces:

$$\sqrt{\frac{F_a}{Db}}\int_0^{\frac{L}{2}}ds = \int_0^{\frac{\pi}{2}}\frac{sin\left(\frac{\delta_0}{2}\right)cos\xi}{\left(\sqrt{1 - sin^2\left(\frac{\delta_0}{2}\right)sin^2\xi}\right)\left(\sqrt{sin^2\left(\frac{\delta_0}{2}\right)cos^2\xi + \frac{\kappa^2 Db}{4F_a}}\right)}d\xi \tag{16}$$

Because the aerodynamic and inertial forces of the actuator are balanced in attached flow conditions, Equation (16) is a good approximation of the quasi-static curvature solution. To account for the deflections of the entire assembled flight control surface, the geometry of the actuator is laid out in greater detail, as presented in Figure 9. The distance from the centerline of the main spar to the end pin is $L_L$, the baseline length of the PBP actuator is $L_o$, and the total distance to the end of the pin is $Lo_{tot}$. One can see that the end pin moves up and down as the PBP element is flexed. This vertical motion in turn rotates the aeroshell around the centerline of the main spar.

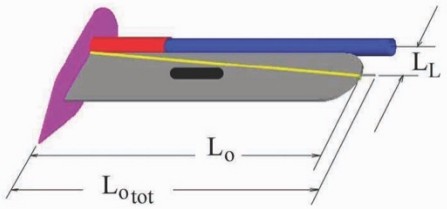

**Figure 9.** Geometry Definitions of Flexspar Actuator Element.

The precompression band was fabricated from a 2 mm dia. silicone band that was pretensioned and affixed to the base. As the element would bend in one direction or another, the precompression band always kept the PBP element loaded in axial compression. As the PBP element is bent, it takes the shell in pitch around the center of the spar, inducing a rotation ($\phi$), as follows:

$$\phi = 2sin^{-1}\left[\frac{1 - cos(\delta_0)}{2\delta_0\left(\frac{L_L}{L_0}\right)} + \left(\frac{L_{0tot}}{2L_L} - \frac{sin(\delta_0)}{2\delta_0\left(\frac{L_L}{L_0}\right)}\right)sin(\delta_0)\right] \quad (17)$$

In addition to general precompression, which places the bending element in a nearly buckled or, with induced imperfections, a controlled post-buckled state, another important form of precompression is used. This piezoelectric element precompression is critical to maintain elemental toughness and allow the highest performance levels to be achieved. Developed more than two decades ago, this method places the substrate in tension while the piezoelectric elements are placed in compression [36]. If one examines total laminate in-plane strain as a function of temperature difference from a reference condition, using the relative in-plane stiffness ratio ($\psi$):

$$\epsilon = \frac{(E_a A_a \alpha_a + E_s A_s \alpha_s)}{E_a A_a + E_s A_s}\Delta T = \frac{(\alpha_a + \psi\alpha_s)}{1 + \psi}\Delta T \quad (18)$$

If one solves for the precompression levels of the piezoelectric elements, then it is easy to see that the difference in linear coefficient of thermal expansion of the actuator and substrate, $\alpha_a$ and $\alpha_s$ respectively, play an important role:

$$\epsilon_a = \epsilon - \alpha_a\Delta T = \frac{(\alpha_a + \psi\alpha_s)}{1 + \psi}\Delta T - \alpha_a\Delta T = \frac{(\alpha_s - \alpha_a)}{1 + \psi}\psi\Delta T \quad (19)$$

The substrate pretension strain levels can be calculated as follows:

$$\epsilon_s = \epsilon - \alpha_s\Delta T = \frac{(\alpha_a + \psi\alpha_s)}{1 + \psi}\Delta T - \alpha_s\Delta T = \frac{(\alpha_a - \alpha_s)}{1 + \psi}\Delta T \quad (20)$$

To properly lay out the design of a PBP actuator, one would determine the maximum curvature possible and the difference between the cure temperature and minimum service temperature. Then, adjust the thickness levels so that the substrate would fail in tension ($f_{ty}$) at that minimum temperature condition, considering a safety margin or factor as well. This design method has been found to harden PBP actuators so well that they can even withstand launch accelerations consistent with artillery shells [36].

While these techniques were specifically intended for use in artillery shells and munitions, a number of investigators applied various forms of precompression and structural instability to drive deflections higher in larger structures like entire sheets and airfoils. Schultz and Hyer established several important snap-through characteristics of composite lamina using cross-ply coupling to thermally induced strains [37]. Giddings et al. also examined a number of bistable composite structures using piezoelectric materials to trigger deflections [38,39]. Multiple modes of stability beyond just bistable configurations were investigated as efforts were underway for developing continuous control of lamina

through piezoelectric triggering mechanisms [40]. The early 2000s also saw the use of thermally induced stresses to amplify piezoelectrically triggered deflections in the private industry from FACE and NASA via the "Thunder" program [41]. Finally, much larger scale airfoil-like structures using twisting mechanisms were explored again by Schultz in a drive for full flight control [42].

## 4. Actuator Element and Flight Control Surface Buildup

The Flexspar PBP actuator element was fabricated with a pair of 127 μm thick PZT-5H elements mounted on either side a 76 μm AISI 1010 stainless steel substrate. The PZT was bonded to the substrate by a 104 μm thick layer of Scotchweld™ epoxy in an elevated temperature cure, as described in Section 3. A film of EP21TDC-N conducting epoxy was included in the bond layer to maintain electrical connectivity between the substrate and piezoelectric element. Figure 10 displays the layup and geometry of the Flexspar PBP actuator element.

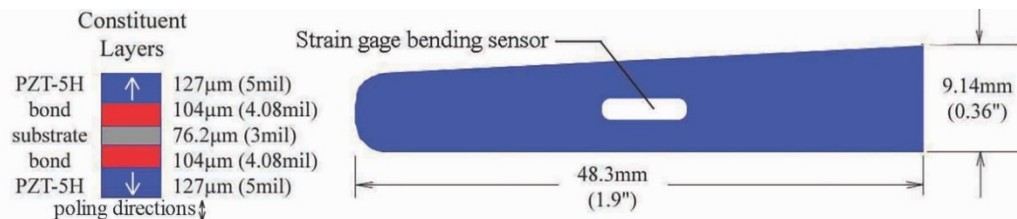

**Figure 10.** PBP Flexspar Element Layup.

With a cure temperature of 350 F (177 °C) and a minimum service temperature of −76 F (−60 °C), the piezoelectric element coefficient of thermal expansion (CTE) precompression was maximized for this design. The actuator was coated with oligomeric silane following assembly to mitigate edge field breakdown. The actuator element was poled in a 5200 V/mm static field at 20 C for 5 min, demonstrating extremely high resistance to breakdown. A full hard AISI 304 drawn stainless main spar and bushing sleeve were mated to a 0.020" (0.51 mm) thick steel base plate via a 2–56 threaded post. A retainer screw at the end of the main spar eliminated spanwise jitter and slop. Figure 11 displays the major constituents of the actuator element along with the aeroshell and main structural components.

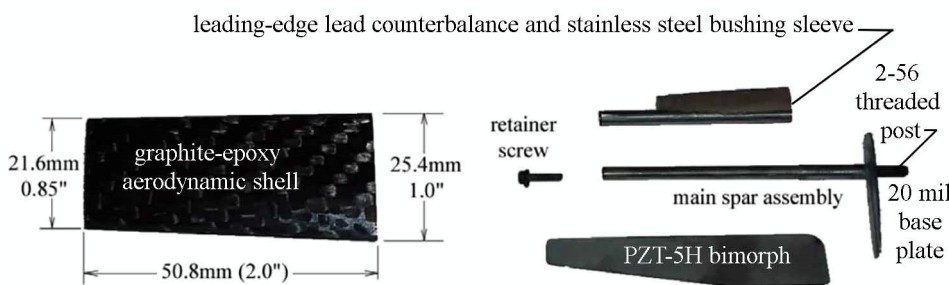

**Figure 11.** Major Components of Flexspar Actuator and Aeroshell.

The components of Figure 11 were joined into an internal actuator assembly, then fitted with a polyurethane precompression band that was mounted to the base and pretensioned to limit buckling tension. The aeroshell was then fitted over the actuator assembly and secured with the retainer screw for a total weight of just 0.18 oz (5.2 g).

## 5. Static and Dynamic Actuator Testing

### 5.1. Test Setup

Static, quasi-static, and dynamic testing of the Flexspar actuator was conducted on a seismic test rig. The test rig was constructed from a solid, seismic mount upon which the

stabilator was mounted. Rotational deflections were measured in two ways: (i) The strain gage pair presented in Figure 12 was used to track bending deflections of the Flexspar element. Because the shell rotation could be related immediately to bending deflection, the two were correlated; (ii) The rotational deflections and bending deflection correlation was measured to within 0.01 deg. resolution from a 1mm square, 1/4 wavelength mirror chip mounted on the leading edge of the stabilator (flooded with green laser light, below, Figure 13). Reflections from the laser were tracked two meters away for accuracy. Dynamic commands and deflections were recorded with a 16 kHz, 16 bit National Instruments PXle-6124 data acquisiton system. Static and dynamic correlation between measured strain gage deflections and observed shell rotations was achieved via repeated testing through 200 Hz. A HiTec HFP-25 was modified to take PWM signals and push them to high voltage command levels compatible with PBP actuators.

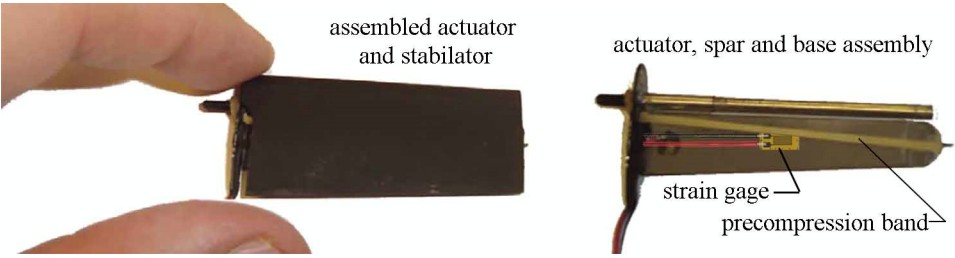

**Figure 12.** Flexspar Actuator Assembly and Flexspar Stabilator.

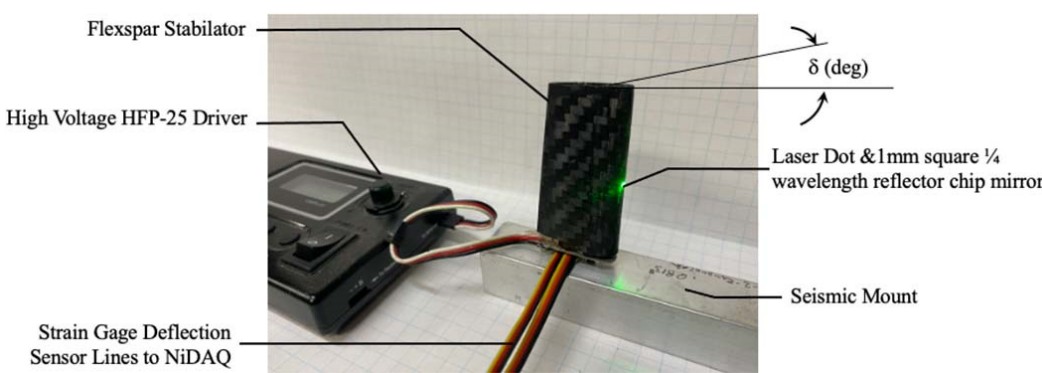

**Figure 13.** Test Setup with Laser Reflection Mirror and 1/4" (6.35 mm) Quadrule.

Testing was conducted at 72 F (22 °C), 29.90–29.98" (759–761 mm) Hg, in 75–89% relative humidity.

*5.2. Test Results*

The first test measured the quasi-static peak-to-peak deflection-moment relationships. Tests were conducted at 20 °C, 100 min after repoling at 2000 V/mm. The actuator generated predictable, regular deflections, matching theory and experiment almost precisely.

From Figure 14, it is clear that the models capture the undeflected root pitching moment behavior well. That said, they overpredict the real actuator performance at high deflection levels. It is thought that nonlinearities in the precompression band induce small deviations. In any case, closing the loop between deflection commanded and deflection generated is easy by using a simple PIV loop with strain gage sensors measuring bending and therefore rotational deflections.

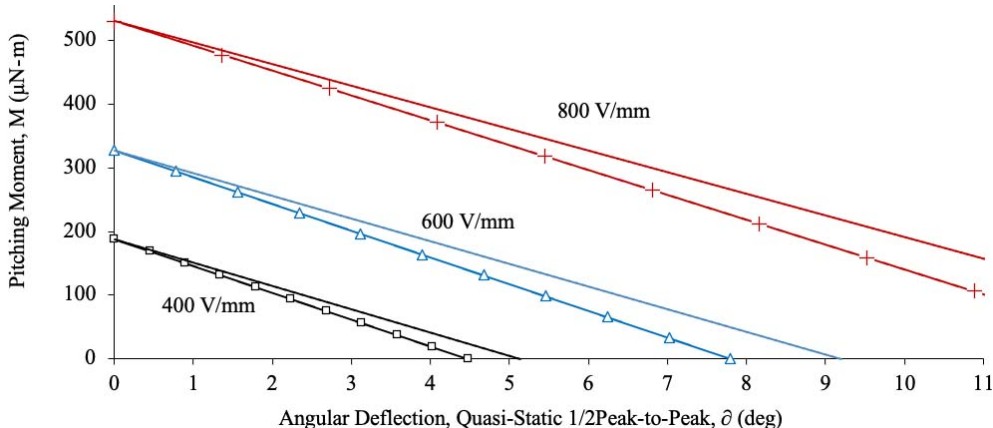

**Figure 14.** Quasi-Static Moment-Deflection Results.

Dynamic testing was conducted using a sinusoidal excitation for the open-loop response. From Figure 15, it easy to see a resonance peak around 22 Hz with a corner frequency of approximately 28 Hz. A Limit Dynamic Driver (LDD) was developed to push the dynamic response to far higher levels. This Limit Driver was designed to overdrive the PZT elements in their poled directions up to the edge breakdown field strengths, while observing tensile limits (governed by temperature constraints). Reverse field strengths going against the poling direction were limited to just 200 V/mm so as to eliminate the risk of depoling. The total peak power consumption measured was under 320 mW at 126 Hz (the pseudo resonance peak) through the 150 Hz corner. The voltage rise rate limit during testing was limited to 8.6 MV/s, as the actuators were driven to breakdown voltage limits. Because edge, atmospheric, and through-thickness breakdown field strengths are highly nonlinear, experimental testing demonstrated them to be in excess of 3200 V/mm under the most severe tensile loads associated with laminate curvature for the actuator configuration given the oligomeric silane surfacant coating. Accordingly, a 10% safety margin was employed during actuation leading to an instantaneous field limit of 2880 V/mm during high rate actuation. It should be noted that the 10% safety margin proved satisfactory for both bench and wind tunnel testing, as no arcing was detected during any of the tests.

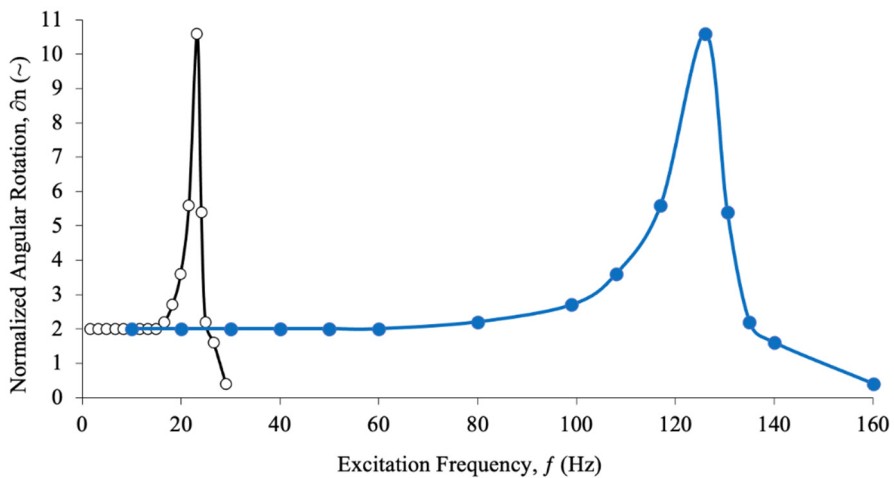

**Figure 15.** Open-Loop and Closed-Loop LDD Deflection Results.

Testing of the stabilator was also conducted in the University of Kansas 12 × 18″ (30 × 45 cm) subsonic wind tunnel. The stabilator was mounted on a 10-chord boundary layer splitter plate. Because the stabilator was balanced around the line of aerodynamic centers, there was no change in quasi-static deflection levels with increasing airspeed.

The dynamic testing did show some rather profound differences as the closed-loop dynamic test frequencies became so high that the reduced frequencies exceeded a value of 1 at low speeds. Accordingly, shed spanwise vortex damping clearly impacted resonance peak maxima. Testing was conducted at 72 F (22 °C) 29.97 inches (761 mm) mercury.

Figure 16 displays the effects of high reduced frequency damping on dynamic deflections. Clearly, aerodamping effects are increased with increasing frequency. While the resonance peak was clearly reduced at resonance, the dynamic corner frequency was still in excess of 140 Hz. Given the aeromechanics explained in the introduction of this paper, a corner frequency in excess of 140 Hz is more than enough to properly capture and control even the fastest aeromechanical modes for the high speed subscale aircraft under consideration. While driving the stabilator at higher deflection levels was certainly possible, as the deflections would breach ±11°, the shell would reach the bump stops, which would prevent rotation levels beyond the rotational limit.

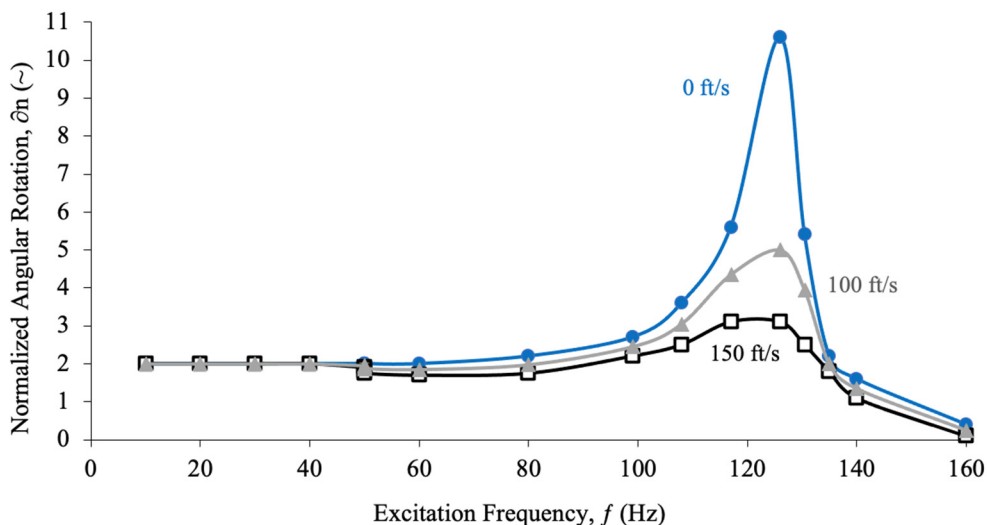

**Figure 16.** LDD Stabilator Deflection Decay with Increasing Airspeed.

## 6. Conclusions

This study has demonstrated that extremely high speed micro actuators for subscale aircraft can be made to so small that they can be packed within the aerodynamic shell of a 1″ (25.4 mm) chord × 2″ (50.8 mm) semispan stabilator, with a total weight of just 0.18 oz (5.2 g). By using post-buckled precompressed (PBP) piezoelectric elements, a 380 mg mass actuator element can drive the aeroshell at high rates, up to a 26 Hz corner frequency. By combining laminated plate theory with a buckling analysis based on axial precompression levels along with rotational kinematic relationships, open-loop rotational deflections and moment generation can be predicted within a few percent. The models were found to overpredict open-loop performance at high deflections approaching ±11°. By using an actuator-based feedback loop, commanded and measured deflections could be made to match within 0.01°. By employing the coefficient of thermal expansion (CTE) laminate precompression of the piezoelectric elements, they could be safely driven to extremely high levels, exceeding 2000 V/mm. Using a Limit Dynamic Driver (LDD) which could apply such high fields for short durations, a pseudo-corner frequency in excess of 140 Hz (880 rad/s) was achieved, which is well in excess of the control speeds needed to control even the highest frequency aeromechanical modes of high speed micro drones.

**Author Contributions:** N.W. conceived and developed the aircraft design. R.B.-G. performed the actuator analysis. All authors have read and agreed to the published version of the manuscript.

**Funding:** This research was funded by the University of Kansas Transportation Research Institute (KUTRI).

**Institutional Review Board Statement:** Not applicable.

**Informed Consent Statement:** Not applicable.

**Data Availability Statement:** Supporting data found in KUTRI laboratory notebooks.

**Conflicts of Interest:** The authors declare no apparent or real conflict of interest in the authoring of this document.

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
