# Peer review of "High Speed Microactuators for Low Aspect Ratio High Speed Micro Aircraft Surfaces"

_actuators, doi:10.3390/act10100265_

Round 1
Reviewer 1 Report
Nice paper on piezoelectric micro-actuators for aero-applications. Written clearly and in an engaging style. Part review and new results.
My only point is the citations are all from a relatively narrow range. There is also work on trying to exploit instability or internal stresses via Thunder type piezo-configurations. There should be at least some mention to alternative approaches and why the approach in this paper is more suitable - this will provide more balance to the paper.
e.g.
[1] Schultz MR. A concept for airfoil-like active bistable twisting structures. J Int MaterSyst Struct 2008;19:157–6 and Shultz/Hyer approaches.
[2] Piezo and bistable combinations. Modelling of piezoelectrically actuated bistable compositesPF Giddings et al Materials Letters 2011 V 65, Issue 9, 15 May 2011, Pages 1261-126
[3] Characterisation of actuation properties of piezoelectric bi-stable carbon-fibre laminates P Giddings et al Composites Part A: Applied Science and Manufacturing 39, 697-703 2008
[4] Thunder types approach e.g. Revolutionary Innovations in Piezoelectric Actuators and Transformers at FACEA. Vázquez Carazo*, Department of R&D Engineering, Face Electronics, LC 427 W. 35th Street, Norfolk, Virginia 23508, USA
Nice paper.
Author Response
These are excellent observations. All references have been added and more as Marc Shultz's work in particular is outstanding and Vazquz's FACE/Thunder efforts underpinned adjacent work in full-scale/inhabited aircraft. Verbiage describing these tremendous contributions was also added.
Reviewer 2 Report
This paper provided many specification and parameter for subscale aircraft. It was rarely seen research paper with many of figure from reference. It suggests to be deleted, unless this manuscript is defined as the review article. Almost all equations are lack of the illustration on the parameters. Never rigorous research paper happened this error. Because of no explanation in parameter, reviewer cannot figure out what the key parameters is designed for improvement aircraft. Meanwhile, only schematics and photo show in Figs. 9-12, without the experimental setup. Reviewer doesn't know how to measure the reliable results as shown in Figs. 13-15. After description in detail, the paper can be read easily by anyone.
Author Response
It appears that the reviewer notes that the research paper uses a few figures based on previous work to underpin the technology and justify the approach to the reader. Unfortunately, Figure 2 was mis-attributed to Harasani (Ref. 2), when it is the author's original, unpublished work and unique to this paper. Hopefully this will alleviate some concern. The reviewer is asked to note that the remaining few figures from other sources are important to underpin the discussion. The paper bridges widely disparate technologies between adaptive structures and aeromechanics communities. Readers from both disciplines will be unfamiliar with technology from the other one. Accordingly, a small bit of cross-disciplinary education is appropriate because adaptive structures technologists often know little of aeromechanics and visa versa as noted by the other reviewers of this paper.
The reviewer notes that symbols are shown in some equations, but not shown in a separate figure. In an effort to take advantage of standard nomenclature of the field, expressions like Equation 2 simply call upon that standard nomenclature. The reviewer has asked for graphical illustration of each term in Equation 2. Figure 8 has been updated to clearly show what each geometric term refers to.
The reviewer also commented that although the experimental setup was discussed, it was not shown in a figure; accordingly, a new Figure 13 has been added showing the stabilator in the seismic test rig during dynamic testing.

Reviewer 3 Report
The paper is well-written and interesting. Nevertheless, the following points should be addressed:
- Fig. 1, 2, 5, 6 - If the figure was reproduced the copyrights should be indicated in the description;
- Eq.1 description of the variables used in the eq should be added in the text for a better understanding (this applies to the whole text);
- Fig. 4 - Unclear description - there are 4 quantities listed with one unit (deg) and one quantity shown in the chart;
- L127 - outward?
- L165-166 - The caption of Fig. 6 should describe what A, B and C refer to;
- Eq 2-17 - eq. numbers not aligned properly; format problems, seems that an image was used by the authors instead of the equation;
- L 314 - please use the full names of the units;
- Fig. 13 - the unit of the Moment quantity should be µN⋅m not µN-M
- L341-349 a more detailed description of the LDD structure would be welcomed;
- L360 - at 22° C and 29.97 inHg (if inches of mercury);
- The authors should underline the novelty of their solution compared to the latest advances in the field, especially if the PBP designs mentioned in the introduction are 9 or more years old.
Author Response
The reviewer astutely notes that several figures rely upon data from earlier references:
Reviewer: Fig. 1, 2, 5, 6 - If the figure was reproduced the copyrights should be indicated in the description;
Authors: Fig. 1 uses data from Ref. 2, but the artwork has been completely recast in a new format with new limits showing more clearly the zone of interest for sub-sub microservoactuators.
Fig. 2 was mis-attributed and is the original work of the author.
Fig. 5 uses data from Ref. 22 - 28 and 32, but has been modified for clarity showing how the PBP configuration moves transfer efficiencies from lower levels to greater than 90%. The original data shows no such PBP efficiency enhancement.
Fig. 6 copyright is held by the principal author. Accordingly, the principal author hereby gives permission to MDPI to use the figure.
Reviewer: Eq.1 description of the variables used in the eq should be added in the text for a better understanding (this applies to the whole text);
Authors: Figure 8 has been modified to clearly show the geometry of the PBP element, showing terms of Equation 1.
Reviewer: Fig. 4 - Unclear description - there are 4 quantities listed with one unit (deg) and one quantity shown in the chart;
Authors: The principal rotational error measured in Fig. 4 is actuator slop. The axis title has been changed accordingly.
Reviewer: L127 - outward?
Authors: "aftward" is correct.
Reviewer: L165-166 - The caption of Fig. 6 should describe what A, B and C refer to;
Authors: Caption modified to describe in words points A, B, and C.
Reviewer: Eq 2-17 - eq. numbers not aligned properly; format problems, seems that an image was used by the authors instead of the equation;
Authors: Copy editor is asked to help with this final formatting.
Reviewer: L 314 - please use the full names of the units;
Authors: Done
Reviewer: Fig. 13 - the unit of the Moment quantity should be µN⋅m not µN-M
Authors: Done
Reviewer: L341-349 a more detailed description of the LDD structure would be welcomed;
Authors: Done -- 111 words describing the limits applied and structure of the LDD.
Reviewer: L360 - at 22° C and 29.97 inHg (if inches of mercury);
Authors: Units spelled out in both English and SI
Reviewer: The authors should underline the novelty of their solution compared to the latest advances in the field, especially if the PBP designs mentioned in the introduction are 9 or more years old.
Authors: Verbiage added in the Abstract describing the multifold increase in dynamic response.

Round 2
Reviewer 2 Report
The revision provided complementary to make reader understanding more information from this paper. Although the study enriches not enough research, the review information might give researcher some new idea.